# Comment on Ibrahim et al. The Effects of In-Plane Spatial Resolution on CT-Based Radiomic Features’ Stability with and without ComBat Harmonization. *Cancers* 2021, *13*, 1848

**DOI:** 10.3390/cancers13123037

**Published:** 2021-06-18

**Authors:** Fanny Orlhac, Irène Buvat

**Affiliations:** Laboratory of Translational Imaging in Oncology, U1288 Inserm, Institut Curie, Université Paris Saclay, 91401 Orsay, France; irene.buvat@u-psud.fr

We read with great interest the paper by Ibrahim et al. [1], dealing with the harmonization of radiomic feature values extracted from CT images using the ComBat method. Using phantom data, the authors studied the effect of 10 different image resampling methods and ComBat harmonization on the reproducibility of radiomic features. They concluded stating the superiority of resampling methods over harmonization using ComBat. However, we would like to discuss some points in the study that we believe may mislead users of ComBat.

The authors used the publicly available Credence Cartridge Radiomics (CCR) phantom data [2] composed of 10 different texture patterns. To evaluate the effect of in-plane spatial resolution on radiomic feature values, the authors used CT images from this phantom scanned using two scanner models with various pixel spacing for each. They segmented a single volume of interest (VOI) in each layer of the phantom (i.e., 10 VOIs per scan) and calculated 91 handcrafted radiomic features using Pyradiomics [3] from each VOI. For example, between CCR-2-001 and CCR-2-007 scans that differ only in pixel spacing (0.39 × 0.39 mm^2^ versus 0.98 × 0.98 mm^2^), the authors showed that only 39/91 (42.9%) features lead to a concordance correlation coefficient (CCC) greater than 0.9 before ComBat. After ComBat harmonization performed by pooling all 10 VOIs of each scan in the same batch, the number of radiomic features with CCC > 0.9 only increased to 48/91 (52.7%) features.

This result is not satisfactory since only half of the features were correctly harmonized, and this finding is contrary to what has been reported in the literature using the same [4] or a different phantom [5]. We believe that this is due to a misuse of ComBat. Indeed, a central assumption of ComBat is that all measurements grouped in the same batch are equally affected by the imaging protocol. As Mackin et al. [6] have already shown, not all layers of the CCR phantom, corresponding to different textural patterns, are affected in the same way by the imaging protocols, so all measurements cannot be harmonized with the same ComBat transformation.

We re-analyzed the data used by Ibrahim et al. [1], focusing on the CCR-2-001 and CCR-2-007 scans. For each of them, we segmented 16 spherical VOIs of fixed size (4 mL) in each layer (160 VOIs in total). Indeed, a specific ComBat transformation must be estimated for each layer; thus, several VOIs from each layer are needed to determine each layer-specific transformation. We calculated 93 available radiomic features with Pyradiomics (version 3.0.1) using the same specification as in the paper (bin size = 25 HU, no spatial resampling, no shape feature), as we had no information about which 2 features were not present in the paper by Ibrahim et al. [1] that mentions 91 features only. We used the same evaluation metrics (CCC over all layers) as in the original paper (cf. Tables 1–5 of [1]). Before ComBat, we showed that 36/93 (38.7%) features lead to a CCC greater than 0.9, which is consistent with [1]. When we harmonized all of them with the same ComBat transformation, as performed by the authors but which is not correct based on ComBat assumptions, only 64/93 (68.8%) features were concordant between the two scans (CCC > 0.9). When we used ComBat separately for each layer, in agreement with the ComBat requirements, 80/93 (86%) features were concordant (CCC > 0.9), and only 4 features resulted in a CCC lower than 0.8. We obtained similar results with another Image Biomarker Standardisation Initiative (IBSI)-compliant [7] software, LIFEx (version 7.0.0, www.lifexsoft.org (accessed on 20 April 2021)) [8], with 22/44 (50%) features obtaining CCC > 0.9 before data processing and 40/44 (90.9%) features with a CCC > 0.9 after ComBat applied separately for each layer. This demonstrates that unlike what was suspected by the authors in the discussion of [1], the performance of ComBat does not significantly depend on the software used to extract the radiomic features.

We confirmed these results visually by projecting the data in the space spanned by the first two principal components of principal component analysis of the 93-feature vector associated with each one VOI. Figure 1a shows the dispersion of the feature values measured on the two scans, where each color corresponds to a layer, i.e., to a specific textural pattern. The same data are shown in Figure 1b by highlighting the scan (in red: CCR-2-001, in blue: CCR-2-007), where the misalignment between the two scans can be clearly seen for all layers. If a single ComBat transformation is calculated for all layers, the alignment is satisfactory in some patterns but not in others (Figure 1c), because the patterns are not influenced the same way by the pixel spacing. Conversely, the feature values are correctly aligned between the two scans when the ComBat transformation is determined separately for each layer (Figure 1d).

To meet the assumptions of the method, ComBat must be applied separately to each texture pattern. This means that the ComBat transformation can be different between two tissue types, like liver and tumor tissues, for example. It is thus recommended to always scrutinize the feature value distribution in each tissue type and each batch to check that ComBat assumptions are met. In addition, if the characteristics of the patients are different between two centers, for example, with a larger proportion of advanced tumors in one of the centers compared to the others, this should be taken into account in ComBat by introducing a covariate as has already been explained and illustrated previously [9,10].

The second point we wanted to discuss concerns Figure 1 in the paper by Ibrahim et al. [1]. This figure may suggest to ComBat users that data alignment between multiple imaging protocols can be inferred from phantom data and applied to patient data. For the same reasons as for the different phantom layers, a transformation determined on phantom data is not appropriate for patient data. ComBat is a data-driven method and does not require phantom acquisitions to determine the transformations to apply.

In conclusion, we consider that there was a misuse of ComBat harmonization in this paper, which does not allow for a reliable assessment of the respective contributions of the spatial resampling and ComBat on CT-based radiomic features.

## Figures and Tables

**Figure 1 cancers-13-03037-f001:**
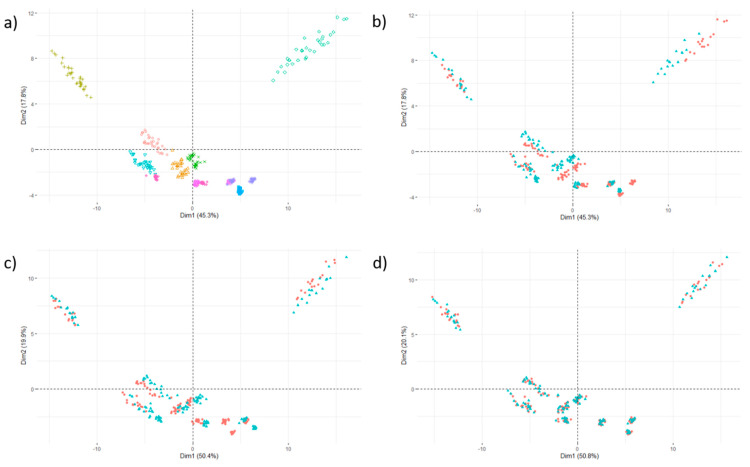
Principal component scores for 160 samples corresponding to 16 volumes of interest × 10 texture patterns represented by 10 different colors (see (**a**)); (**b**) before ComBat (in red: CCR-2-001 scan; in blue: CCR-2-007 scan); (**c**) after ComBat with the same transformation for all texture patterns; (**d**) after ComBat applied separately for each phantom layer. After applying ComBat correctly, the data of the two scans are correctly aligned for all texture patterns.

## Data Availability

The data used in this study are openly available on TCIA.org at https://wiki.cancerimagingarchive.net/pages/viewpage.action?pageId=39879218#3987921801d205cc64754cd2b5e249083c8ae80d (accessed on 20 April 2021).

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
