# Peer review of "Comment on Ibrahim et al. The Effects of In-Plane Spatial Resolution on CT-Based Radiomic Features’ Stability with and without ComBat Harmonization. Cancers 2021, 13, 1848"

_cancers, 2021, doi:10.3390/cancers13123037_

Round 1
Reviewer 1 Report
This is a comment on the recently published manuscript “The Effects of In-Plane Spatial Resolution on CT-Based Radiomic Features’ Stability with and without ComBat Harmonization” by Ibrahim et al. Here the authors pointed out a misuse of the ComBat procedure for radiomic features harmonization.
I think that this comment is pertinent, and it is sustained by the analysis carried out on the same phantom used by Ibrahim et al. The authors demonstrate a solid experience on the use of this specific method, and I believe that this specific comment can be useful for the scientific community, as an explicit indication on how to correctly use the ComBat procedure on radiomic features.
Maybe, the authors could justify their choice of extracting features from 16 VOIs on each layer, instead of the single one used in the manuscript of Ibrahim et al. and how the different VOI volumes impact the final results. If I have correctly understood, this is the only difference between the two works.
About the last point on the use of isotropic voxels, I don’t completely agree with the authors. In the IBSI recommendations, it was stated that 2D interpolation may be beneficial in case the spacing between slices is large compared to the desired voxel size, and/or compared to the in-plane spacing. In addition, in this specific case, each layer represents a different texture, thus I believe that texture features should be computed in-plane without the need of an isotropic voxel. For this reason, I suggest removing this sentence or better justifying it.
Author Response
We thank the reviewer for his/her comments. Indeed, the only difference between Ibrahim et al work and ours (except the way ComBat is applied) was to segment 16 VOIs per layer instead of 1. Several VOIs per layer are absolutely needed to apply ComBat using these phantom data because ComBat needs several samples from the same material to estimate the realignment parameters between the feature values obtained using different pixel spacing. It is impossible to derive a transformation between two statistical distributions if each is described by only one sample. In a cohort of patients, if we had to realign measurements made in the liver, for instance, between two imaging protocols, one VOI per patient would be sufficient as we would have several patients scanned with each protocol. Yet, with the phantom data used by Ibrahim et al, there was a single acquisition per setting (pixel spacing), thus several VOI per material are needed. We now better explain this point in the comment.
Concerning the use of isotropic voxels we understand the point of view by the reviewer and given that this is not the topic of our comment, we removed the sentence.
Reviewer 2 Report
This Comment highlights some crucial points in the application of ComBat harmonization for radiomic analysis. The Authors criticize the approach of [1], where (a) all CCR layers underwent the same ComBat transformation; (b) Fig. 1 and some discussions may be misleading.
In my view, this Comment is highly interesting for the readers. I have some remarks/suggestions:
- The presented analysis has not the same number of features and VOIs as in [1]. Therefore, results are meaningful but not directly comparable. This fact should be mentioned. The adopted radiomic features and VOIs should be reported.
- If the ComBat transformation should be applied to each (known) texture pattern separately, as it is clearly explained by the Authors, one may argue that ComBat is not applicable if actual texture/tissue properties are not exactly the same among all the considered batches. The differences among CCR phantom layers are well-defined and evident, but this may also represent a problem when there are unknown "classes" (e.g. patient groups) that may reflect different pathologic states or physiological variations, and therefore may present some texture/tissue differences. This can happen in radiomic predictive modeling. Some comments about this point will be appreciated.
- In line 51, I suggest writing "the performance of ComBat does not significantly depend on the software" since some differences can be noticed.
- A small typo: line 37, "spherical".
Author Response
We thank the reviewer for his/her comments. Several VOIs per layer are absolutely needed to apply ComBat using these phantom data because ComBat needs several samples from the same material to estimate the realignment parameters between the feature values obtained using different pixel spacing. It is impossible to derive a transformation between two statistical distributions if each is described by only one sample. In a cohort of patients, if we had to realign measurements made in the liver, for instance, between two imaging protocols, one VOI per patient would be sufficient as we would have several patients scanned with each protocol. Yet, with the phantom data used by Ibrahim et al, there was a single acquisition per setting (pixel spacing), thus several VOI per material are needed. We now better explain this point in the comment.
In their paper, the authors do not provide the exact list of features computed with Pyradiomics, we therefore included all 93 features that were available and did not correspond to shape features (shape features are not relevant as all VOI had the same shape). So we have two extra features compared to what the authors reported but this does not alter the findings as the feature realignment is performed for each feature independently.
As perfectly understood by the reviewer, ComBat must be applied separately for each type of tissue or tumor as they may be affected differently by the batch effect. This is why it is recommended to always scrutinize the feature value distribution in each tissue type and each batch to check that ComBat assumptions are met. Also, when the characteristics of the patients are different between batches (e.g. a higher proportion of advanced stages in one of the centers than in the others), this can be taken into account by including a covariate in ComBat (cf. [9]: Orlhac et al. J Nucl Med 2018 and [10]: Orlhac et al. Eur Rad 2021). We have added this point in the comment.
As suggested by the reviewer, we added the term "significantly" and corrected the typography for “spherical”.
Reviewer 3 Report
The authors of this comment state that Ibrahim et al. [1] in their work used incorrectly ComBat to harmonize the radiomic features. The rationale that raised this comment is valid and well placed. The basic consideration, in order to use ComBat correctly, is that the various measurements must have the same dependence on the acquisition protocol. In Mackin et al. [6] it has been shown that with the CCR phantom used in Ibrahim et al. [1] the previous assumption is not respected, so all the measurements cannot be processed with a single transformation, but each slice of the volume must be processed with its own transformation. To validate their statement, authors took the same data and carried out the same elaborations, obtaining a greater number of correlated features (80/93, 86%) than the work of Ibrahim, et al. (48/91, 52.7%). This misuse of ComBat has resulted in a much smaller number of related features: for this reason the study of Ibrahim, et al. seems not to be well placed, not allowing to conclude on its usefulness to address the spatial resampling effects on CT-based radiomic features. If possible, in addition to the work of Mackin et al. [6] where it has been shown that the prerequisite is not respected, it could be useful to insert a reference to the ComBat documentation in which its correct use is highlighted.Author Response
We thank the reviewer for his/her comments. The proper use of ComBat has been previously described and illustrated using examples in references 9 and 10, that we now cite as they may allow readers to better understand the principles and underlying assumptions of the method.
Round 2
Reviewer 2 Report
I would like to thank the Authors for responding to my questions. I am satisfied with the revised version of this Comment.